# Wound Healing, Antioxidant, and Antiviral Properties of Bioactive Polysaccharides of Microalgae Strains Isolated from Greek Coastal Lagoons

**DOI:** 10.3390/md23020077

**Published:** 2025-02-10

**Authors:** Gabriel Vasilakis, Sofia Marka, Alexandros Ntzouvaras, Maria-Eleftheria Zografaki, Eirini Kyriakopoulou, Katerina I. Kalliampakou, Georgios Bekiaris, Evangelos Korakidis, Niki Papageorgiou, Stefania Christofi, Niki Vassilaki, Georgia Moschopoulou, Ioannis Tzovenis, Athena Economou-Amilli, Seraphim Papanikolaou, Emmanouil Flemetakis

**Affiliations:** 1Laboratory of Food Microbiology and Biotechnology, Department of Food Science and Human Nutrition, Agricultural University of Athens, 11855 Athens, Greece; vasilakis.gavriil@gmail.com (G.V.); giorgosbekiaris@yahoo.gr (G.B.); christofistefania@gmail.com (S.C.); spapanik@aua.gr (S.P.); 2Laboratory of Molecular Biology, Department of Biotechnology, Agricultural University of Athens, 11855 Athens, Greece; smarka@aua.gr (S.M.); alexntzouv@gmail.com (A.N.); mzografaki@aua.gr (M.-E.Z.); kalliamp@yahoo.gr (K.I.K.); nikipapageorgiou99@gmail.com (N.P.); 3Laboratory of Cell Technology, Department of Biotechnology, Agricultural University of Athens, 11855 Athens, Greece; geo_mos@aua.gr; 4Sector of Ecology & Systematics, Department of Biology, National and Kapodistrian University of Athens, 15784 Athens, Greece; itzoveni@biol.uoa.gr (I.T.); aamilli@biol.uoa.gr (A.E.-A.); 5Laboratory of Molecular Virology, Hellenic Pasteur Institute, 11521 Athens, Greece; eirhnh63@gmail.com (E.K.); korakidis13@gmail.com (E.K.); nikiv@pasteur.gr (N.V.)

**Keywords:** microalgae, bioactive polysaccharides, wound healing, antioxidants, antiviral properties, *Tetraselmis*

## Abstract

Microalgae have gained significant attention as sustainable sources of high value compounds, such as bioactive polysaccharides that are usually rich in sulfated groups and exhibit antioxidant properties. Here, 14 new microalgae strains of the genera *Tetraselmis*, *Dunaliella,* and *Nannochloropsis*, isolated from Greek coastal lagoons were analyzed to quantify and characterize their polysaccharide content. Heterogeneity was observed regarding the content of their total sugars (5.5–40.9 g/100 g dry biomass). The strains with a total sugar content above 20% were analyzed concerning the content of total, *α*- and *β*-glucans. *Tetraselmis verrucosa* f. *rubens* PLA1-2 and *T*. *suecica* T3-1 were rich in *β*-glucans (11%, and 8.1%, respectively). The polysaccharides of the two *Tetraselmis* strains were isolated and they were mainly composed of glucose and galactose. The isolated polysaccharides were fractionated using ion-exchange-chromatography. The anionic fraction from *T*. *verrucosa* f. *rubens* PLA1-2 was rich in sulfated polysaccharides, had antioxidant capacity, and exhibited healing properties. The anionic polysaccharides from the two *Tetraselmis* strains did not negatively influence the viability of human cells, while exhibiting antiviral properties against the replication of Hepatitis C Virus (HCV), with median efficient concentrations (EC50) at a range of 210–258 μg/mL.

## 1. Introduction

Microalgae have gained significant scientific and industrial attention in recent years as alternative and sustainable sources of value-added metabolites, such as polysaccharides, lipids, proteins, vitamins and biopigments, due to their nutritional and pharmaceutical properties [1,2,3,4,5,6,7,8,9,10,11,12]. In parallel, the advantage of microalgae is the ease of cultivation, growth, and harvesting, independently of climatic or seasonal conditions [13].

Microalgal polysaccharides content varies between 15% and 75% across different species and are classified as intracellular polysaccharides, containing both the storage substances and the structural components of the cell walls, cell-bounded extracellular polysaccharides and released extracellular polysaccharides [14,15]. Both the production and composition of polysaccharides depend on the microalgal strain and are influenced by the nutritional and physicochemical conditions of the cultivation [16]. Polysaccharides present a diverse range of structures, determined by the composition of monosaccharides (40–50 hexoses, pentoses, and uronic acids), branches, and associated additives (such as sulfate groups, acyl groups, amino acids, or fatty acids) [17]. Bioactive polysaccharides, such as sulfated polysaccharides, beta-glucans (*β*-glucans), galactans, etc., are valuable in the food, feed, pharmaceutical, cosmetic, biomedical, and hydrocolloid industries, due to their hypolipidemic and hypoglycemic effects, as well as their antimicrobial, antiviral, antifungal, anti-inflammatory, antioxidant, and anticancer properties. These properties render them useful in various applications, owing to their biocompatibility, biodegradability, and non-toxicity [13,17,18,19,20,21,22,23,24,25].

Skin reconstruction and hydration products represent valuable assets in the field of cosmetology and pharmaceutical industries. Wound healing is a complex process involving tissue repair through several stages, including hemostasis, inflammation, proliferation, and remodeling. This process engages various cells and signaling molecules to regulate the cellular response and restore the structure of the injured tissue. Bioactive polysaccharides could contribute to the healing of wounds, especially in cases of extensive wounds or in patients with chronic wounds, that have difficult healing. Polysaccharides create an optimal environment for cell adhesion and proliferation by interacting with skin proteins, forming a barrier, which retains local moisture. Simultaneously, they exhibit antimicrobial and antiviral effects by stabilizing the skin’s microflora, thus preventing infections and limiting reactive oxygen species (ROS) and inflammation in the wound. Otherwise, the unchecked inflammation could potentially lead to neoplastic differentiation [25,26].

In the present study, 14 newly isolated strains of microalgae from Greek coastal lagoons were initially evaluated for their polysaccharide content. Enzymatic determination of *β*-glucans content and subsequent polysaccharide extraction were performed on selected strains. After fractionation, the recovered eluates were characterized regarding their total sugars, sulfates concentration, and antioxidant capacity. Finally, cytotoxicity, antiviral activity, and wound healing properties of selected eluates were assessed.

## 2. Results

### 2.1. The Total Sugar and β-Glucan Contents Differ Significantly Among Tetraselmis Species

Polysaccharides are composed of sugar monomers; thus, we first determined the content of total sugars in the dry biomass (DB) of each microalgae strain. More information regarding the initial and final cell densities of the cultures, the maximum growth rates, and the final dry biomass concentration values after 10 days of growth is provided in the Appendix A. As shown in Table 1, the content of total sugars differed significantly among the different microalgae strains, with specific species of *Tetraselmis* exhibiting the highest and the lowest levels, respectively. Indeed, in the DB of *Tetraselmis verrucosa* f. *rubens* PLA1-2 the sugar content reached 40.9%, whereas the respective content in the DB of *Tetraselmis* spp. EST1-2 was only 5.5%.

The species exhibiting a total sugar amount in their DB above 20% (*w*/*w*) were further processed to determine the amounts of total glucans and alpha-glucans (*α*-glucans) and these were used to quantify the content of *β*-glucans, as molecules of increased interest. All microalgae exhibiting a total sugar content higher than 20% and thus further processed, belonged to *Tetraselmis* species. Of these strains, the *T*. *verrucosa* f. *rubens* PLA1-2 had the higher content of total glucans (28.7%, *w*/*w*), *α*-, and *β*-glucans (17.7% and 11.0%, *w*/*w*, respectively) in the DB, followed by *T*. *suecica* T3-1 (total glucans 20.8%, *α*-glucans 12.1%, and β-glucans 8.1%, *w*/*w*, in the DB). In general, we observed that the higher the total sugar content in the DB of microalgae, the higher the content of β-glucans. However, the ratio of β-glucans to total sugars was not constant ranging from 26.9% (*w*/*w*) in the case of *T*. *verrucosa* f. *rubens* PLA1-2 and 24.8% (*w*/*w*) in *T*. *suecica* T3-1 to 5.0% (*w*/*w*) in *Tetraselmis* spp. Mes5.

### 2.2. The Polysaccharides of the β-Glucan Rich Tetraselmis Strains Are Mainly Composed of Glucose and Galactose

To further characterize the polysaccharides of the two *β*-glucan-rich strains of *Tetraselmis* (*T*. *verrucosa* f. *rubens* PLA1-2 and *T*. *suecica* T3-1), these polysaccharides were isolated and hydrolysis was performed, followed by High Performance Liquid Chromatography (HPLC) analysis to detect and quantify their sugar monomers. To isolate the polysaccharides from the DB of microalgae, the lipophilic molecules were extracted using acetone. From the pellet of the defatted microalgal biomass, the water-soluble intracellular, structural, and bound molecules were extracted in pure water (ddH_2_O), via the use of ultrasounds, thermal treatment, and vortexing. The proteins of this extract were denatured using a chloroform/n-butanol solution (Sigma-Aldrich, St. Luis, MO, USA) and from the deproteinated aqueous phase, the polysaccharides were precipitated using cold ethanol (Sigma-Aldrich, St. Luis, MO, USA). After their dissolution in ddH_2_O (4 mg/mL), the isolated polysaccharides underwent thermal acidic hydrolysis, the resulting products were derivatized and subjected to HPLC analysis. Representative HPLC chromatograms from samples containing the hydrolyzed polysaccharides of the two microalgae strains are depicted in Figure 1A,B). The sugar monomers were identified and their relative percentage are shown in Table 2.

Our results showed that glucose was the most abundant sugar in the polysaccharides of both microalgal strains, followed by galactose (galactoglucans). The percentage of glucose in the polysaccharides of *T*. *verrucosa* f. *rubens* PLA1-2 and *T*. *suecica* T3-1 reached 71.9% and 66.4% (*w*/*w*), respectively, and the content of galactose was 19.8% and 20.9% (*w*/*w*) in these two cases, respectively. Thus, the glucose (glucans) content of the DB was approximately 29.4% and 21.6% (*w*/*w*), while the galactose (galactans) content was about 8.1% and 6.8% (*w*/*w*) for *T*. *verrucosa* f. *rubens* PLA1-2 and *T*. suecica T3-1, respectively. In addition to the above sugars, mannose, xylose, rhamnose, and ribose were detected in low quantities (<4%, *w*/*w*). The content of glucose in the above polysaccharides’ extracts agreed with the high levels of glucans identified in the DB of these microalgae strains (see Table 1).

### 2.3. The Anionic Fraction of Polysaccharides from T. verrucosa f. rubens PLA1-2 Contains a Significant Amount of Sulfated Polysaccharides and Exhibits Antioxidant Capacity

To concentrate and separate the sulfated polysaccharides from the total polysaccharides content, the ddH_2_O-dissolved polysaccharides (4 mg/mL) of the two *Tetraselmis* strains were fractionated via ion-exchange chromatography, to retain the negatively charged molecules. For each sample of ddH_2_O-dissolved polysaccharides, four eluates were obtained. Eluate E_1_ was recovered directly after loading the ddH_2_O-dissolved polysaccharides onto the ion-exchange column. Next, the eluate E_2_ was recovered from the column using pure water (ddH_2_O) as eluent. Subsequently, the eluates E_3_ and E_4_ were obtained after loading the column with NaCl solutions (Sigma-Aldrich, St. Louis, MO, USA) using gradually increasing the NaCl concentration (0.2 and 1.0 M, respectively). The eluates were desalted and characterized concerning their total sugar content, their concentration of sulfates, and their antioxidant capacity expressed as Trolox equivalents. The results are displayed in Table 3.

The highest concentration of total sugars (2.06 mg/mL) was observed in the case of E_2_ eluates of the fractionated ddH_2_O-dissolved polysaccharides for both *Tetraselmis* strains. In the E_1_, the solvent was also pure water (ddH_2_O), thus 0.68 mg/mL and 0.59 mg/mL of neutral polysaccharide molecules were from *T*. *verrucosa* f. *rubens* PLA1-2 and *T*. *suecica* T3-1, respectively. The weak and strong anionic polysaccharides of E_3_ and E_4_ eluates were determined as 0.66 mg/mL for both eluates, in the case of *T*. *verrucosa* f. *rubens* PLA1-2 and 1.09 mg/mL and 0.59 mg/mL, respectively, in the case of *T*. *suecica* T3-1. The recovered eluates were characterized concerning the presence of sulfates, while their putative antioxidant capacity was determined by using the Ferric Reducing Antioxidant Power (FRAP) assay. The eluates E_1_, E_2,_ and E_3_ did not contain sulfated polysaccharides, nor did they exhibit antioxidant capacity. Similarly, the E_4_ eluate from *T. suecica* T3-1 had a low content of sulfated polysaccharides (0.014 mg/mL) and did not exhibit antioxidant capacity. On the contrary, the E_4_ eluate of the fractionated ddH_2_O-dissolved polysaccharides from *T*. *verrucosa* f. *rubens* PLA1-2 contained a remarkable concentration of sulfated polysaccharides (0.485 mg/mL) and exhibited antioxidant capacity (3.00 μM Trolox). The presence of sulfated and sugar groups was also confirmed by Fourier transform infrared (FTIR) spectroscopy (see Appendix A). Specifically, the observation of the full range spectrum (Appendix A) allowed the identification of three strong peaks at 3390, 1644, and 667 cm^−1^. These peaks could be assigned to a plethora of bond vibrations such as the -OH and -NH_2_ group vibrations for the former peak or the C=O, C-N, N-H stretching for the peak at 1644 cm^−1^. However, due to the aqueous nature of the polysaccharide extracts, these peaks possibly correspond to water bond vibrations and are more specific to the O-H symmetric and asymmetric stretching and the H-O-H bending for the observed peaks at 3390 and 1644 cm^−1^, respectively [27]. The isolation of the region between 1400 and 900 cm^−1^ (Appendix A), which is less affected by water bond vibrations, allowed the identification of peaks at 1338, 1227, 1123, 1080, 1064, and 996 cm^−1^. The weak peak at 1338 cm^−1^ could be assigned to the asymmetric stretching vibration of COO- of carboxylic acid salts, in combination with the respective symmetric stretching vibration at 1644 cm^−1^. This peak could also be assigned to some proteinic residues (Amide III region; C=N stretching, C=O stretching, N-H bending vibration) [28]. The peak at 1227 cm^−1^ could be associated with the S=O stretching of the sulfated groups in sulfated polysaccharides [29], while the peaks at 1200–900 cm^−1^ are mostly related to polysaccharides and especially the C-O stretching (1123 cm^−1^), the pyranose ring stretching (1080 and 1064 cm^−1^) and the alkene C-H stretching of the ring (996 cm^−1^) [28].

### 2.4. Biological Properties of the Strongly Anionic Polysaccharides of the Two Tetraselmis Strains

The E_4_ eluates (strongly anionic polysaccharides) of both *Tetraselmis* strains that were found to contain high and low levels of sulfated polysaccharides in the case of *T. verrucosa* f. *rubens* PLA1-2 and *T. suecica* T3-1, respectively, were subsequently assessed for effects on the viability of human cells, and for putative wound healing and antiviral properties.

#### 2.4.1. The Strongly Anionic Polysaccharides from the Two Tetraselmis Strains Do Not Negatively Influence the Viability of Human Primary Cells

Normal human dermal fibroblasts (NHDF) cells (primary cells) were used to check for putative effects of E_4_ eluates derived from *Τ. suecica* T3-1 and *T*. *verrucosa* f. *rubens* PLA1-2 strains, on cell viability. The viability of cells was evaluated using the 3-[4,5-dimethylthiazol-2-yl]-2,5 diphenyl tetrazolium bromide (MTT) colorimetric assay. The cells were incubated with a culture medium containing various concentrations of the E_4_ eluates (i.e., 50, 100, 250, and 500 μg/mL) for 24 and 48 h. Control cells were incubated with a culture medium supplemented with the respective volume of ddH_2_O. The MTT assay was performed at the end of the incubation period and the results are presented in Figure 2.

As shown in Figure 2, the viability of the normal human primary cells was not negatively affected by the 48-h incubation of cells with the E_4_ eluates, even when a concentration of 500 μg anionic polysaccharides per mL of cell culture medium was used. Actually, in several cases, the presence of the anionic polysaccharides had a slight positive effect on the cell viability, compared to the control cells.

#### 2.4.2. The Strongly Anionic Polysaccharides from the *T. verrucosa* f. *rubens* PLA1-2 Strain Exhibit Wound Healing Properties

As cell migration governs the wound healing process to restore the integrity and function of the injured tissue, an in vitro assay using normal human dermal fibroblasts (NHDF) cells was performed to simulate the wound healing procedure [30], in order to test the E_4_ eluates from *T. verrucosa* f. *rubens* PLA1-2 and *Τ. suecica* T3-1 for potential wound healing properties. In brief, NHDF cells seeded in wells were allowed to reach a confluency of almost 80%. Then, a scratch was made in the layer of cells and the migration of cells to close the produced gap, in the presence (or not, control cells) of different concentrations of the strongly anionic polysaccharides of the two *Tetraselmis* strains, was monitored. The closure of the gap due to the migration of cells simulated the wound healing procedure. Two concentrations, 300 μg and 500 μg of the strongly anionic polysaccharides per mL of cell culture medium were tested. The cells used as control were incubated with a culture medium supplemented with the respective volume of ddH_2_O. The results are presented in Figure 3.

The E_4_ eluate from *Τ. suecica* T3-1 did not affect the wound healing procedure negatively but also did not exhibit any wound healing properties, neither at 300 nor at 500 μg/mL, compared to the control (approximately 20% relative wound density) after 48 h of incubation. A significant positive effect on the simulated wound healing procedure was observed when the cells were treated with the E_4_ eluate from *T. verrucosa* f. *rubens* PLA1-2, at both concentrations used. About 58% relative wound density was achieved when a concentration of 500 μg/mL was applied, while 44% relative wound density was observed in the case of 300 μg/mL. This positive effect was also dose-dependent, as the use of a higher concentration of the E_4_ eluate from *T. verrucosa* f. *rubens* PLA1-2 resulted in greater wound density.

#### 2.4.3. The Strongly Anionic Polysaccharides from the Two Tetraselmis Strains Exhibit Antiviral Activity Against the Replication of Hepatitis C Virus (HCV)

In this study, we tested the E_4_ eluates from *T. verrucosa* f. *rubens* PLA1-2 and *Τ. suecica* T3-1 for antiviral properties against the Hepatitis C Virus (HCV). To test the E_4_ eluates for potential effects on HCV replication, we used the human hepatoma Huh5-2 cell line that stably harbors the subgenomic reporter replicon of the HCV genotype 1b (Con1 strain) [31]. The HCV genotype 1 and especially the subtype 1b is the most prevalent group worldwide. This HCV replicon expresses the reporter protein firefly luciferase (F-Luc) the activity of which is directly correlated with the replication level of the replicon, and it has been used extensively for the testing/development of new treatments for HCV [32].

To check for effects on HCV replication by the E_4_ eluates from *T. verrucosa* f. *rubens* PLA1-2 and *Τ. suecica* T3-1, the Huh5-2 cells harboring the HCV reporter replicon, were treated for 72 h with the indicated concentrations (i.e., 50, 100, and 200 μg of the strongly anionic polysaccharides per mL of cell culture medium) of these Ε_4_ eluates. The cells used as control were incubated with a culture medium supplemented with the respective volume of ddH_2_O. Then, the cells were lysed, and the viral RNA replication-derived firefly luciferase activity was determined and normalized to the cell’s total protein amount. The results are presented in Figure 4A,B.

As shown, both microalgal E4 eluates exhibited dose-dependent antiviral activity against the HCV replicon without affecting the viability of Huh5-2 cells. Specifically, in the higher concentrations tested (i.e., 100 and 200 μg/mL), the strongly anionic polysaccharides from the *Tetraselmis* strains reduced the HCV replication with the highest inhibition (more than 40%) observed in the case of 200 μg/mL. The median effective concentration (EC_50_) of the strongly anionic polysaccharides of E4 eluates, from *Τ*. *suecica* T3-1 and *T*. *verrucosa* f. *rubens* PLA1-2 strains, was calculated at 257.3 μg/mL and 210.8 μg/mL, respectively.

## 3. Discussion

Polysaccharides isolated from microalgae have been shown to exhibit many desirable properties and have applications in the food industry, medicine, and cosmetology. The synthesis and composition of polysaccharides depend upon the microalgae strain and the conditions of cultivation such as nutrient availability, light quality and intensity, salinity, and temperature. Usually, stress conditions induce the production of polysaccharides but at the same time compromise the growth of microalgae [33]. *Tetraselmis* species have been shown to produce large amounts of bioactive polysaccharides [24]. In the present study, many *Tetraselmis* species initially isolated from Greek coastal lagoons, were included to select the ones that produce high amounts of polysaccharides under standard conditions of cultivation. To do so, the different microalgae strains were grown under standard conditions and the microalgae dry biomass (DB) was characterized concerning the total sugar content and the presence of *α*- and *β*-glucans.

Interestingly, a high level of sugars in the DB of *Tetraselmis verrucosa* f. *rubens* PLA1-2 was achieved using a cultivation medium (Walne’s solution of adjusted salinity 40‰) which was not supplemented with carbohydrates. Similar levels of sugars have been observed for *Arthrospira platensis* (41.1%), *Chlorella vulgaris* (47.9%), and *Dunaliella salina* (31.7%) species but only when their cultivation medium was supplemented with carbohydrates (0.05 g/L) [34]. This stain exhibited also the highest amount of α- and b-glucans in the DB. α-glucans, produced by microalgae as exopolysaccharides or found as wall components or as energy storage compounds, have been shown to act as prebiotics and to exhibit antiproliferative and immunostimulatory properties [35]. *β*-glucans, found in microalgae as storage compounds, cell-bounded, or as wall components, can positively affect human metabolism due to their antioxidant, hypoglycemic, and hypocholesterolemic properties, while can also modulate the immune responses acting as immunostimulants, anti-inflammatory and anti-cancer agents [36,37].

Previous studies comparing several microalgae species, with respect to *β*-glucan content in the DB, had revealed that *Scenedesmus ovalternus* 52.80, *Porphyridium purpureum* 1380-1d, *Cylindrotheca fusiformis* 1017/2, and *Tetraselmis suecica* 66/22c strains produced very high amounts of *β*-glucans (24.2%, 22.4%, 17.9%, and 16.1%, respectively) under standard conditions of cultivation, whereas most of the rest microalgae species studied showed a *β*-glucan content between 5 and 10% of DB [38]. In the same study, several of the above *β*-glucans-rich microalgae strains exhibited also very high levels of total carbohydrates in their DB, with *Porphyridium purpureum* 1380-1d reaching 59.2%, while the maximum content of α-glucans was up to 12% and was observed in the same strain. Based on the above, in our study, the maximum amounts of α-glucans (17.7%) and *β*-glucans (11.0%) in the DB, detected in the case *T. verrucosa* f. *rubens* PLA1-2, represented values close to the typical ones for most microalgae strains.

The composition of the polysaccharides derived from *T*. *suecica* T3-1 and *T*. *verrucosa* f. *rubens* PLA1-2 consisted of a significant proportion of galactose, in addition to glucans. Previous studies involving the microalgae species *Chlorella vulgaris*, *Phaeodactylum tricornutum*, *Porphyridium purpureum*, *Scenedesmus ovalternus*, *Nannochloropsis salina*, and *Dunaliella salina* have revealed that the sugars detected in our *Tetraselmis* strains are often found in polysaccharide extracts of microalgae, glucose and galactose are usually the most abundant sugars detected, and their percentage can vary considerably between microalgal species [13,39].

Galactan polymers and especially their sulfated forms have attracted scientific interest, due to their various properties, such as rheological, antimicrobial, antioxidant, anticoagulant, antithrombotic, and anticancer activities. These properties are directly linked to the structure of the compounds. Typically, galactans are sourced from plants and algae, particularly macroalgae and specifically Rhodophyta, they have a high molecular weight (>100 kDa), demonstrating significant biological activity as hydrocolloids and exhibit distinctive rheological behavior. They are commonly used as stabilizers in various food products, including dairy, bakery, confectionery, etc. [40,41]. Studies have also demonstrated that sulfated carrageenans possess immunomodulatory properties and antitumor activity against Ehrlich-Lettre ascitic carcinoma (EAC) and breast adenocarcinoma [40,42].

Moreover, in our study, the polysaccharides’ extracts of *T*. *verrucosa* f. *rubens* PLA1-2 and *T*. *suecica* T3-1 were fractionated and checked for the presence of sulfates and for putative antioxidant properties. As in the E_1_ and E_2_ eluates, the eluent was pure water (ddH_2_O), these fractions were rich in neutral polysaccharides that can not bind to the positively charged column. The weak and strong anionic polysaccharides were recovered at the E_3_ and E_4_ eluates, respectively. Based on the results, the amount of the anionic polysaccharides is lower than that of the neutral polysaccharides in both *Tetraselmis* strains. All polysaccharides’ eluates were characterized concerning the presence of sulfates and their putative antioxidant capacity and only the E_4_ eluate of *T*. *verrucosa* f. *rubens* PLA1-2 exhibited high content of sulfated polysaccharides and antioxidant activity. The water-soluble polysaccharides from *Tetraselmis* spp. strains have been shown to present significant antioxidant, antifungal, and tyrosinase inhibitory activities [24]. Moreover, sulfated polysaccharides from microalgae have attracted interest due to their medicinal properties. Sulfated polysaccharides are found mainly in marine organisms, specifically as a key component of cell walls and have significant physiological roles, such as helping to retain moisture protecting the microalgae against drying or acting as a protective barrier against pathogens [43]. Sulfated polysaccharides encompass a diverse group of anionic polymers of high pharmaceutical interest. Actually, the sulfated polysaccharides isolated from *T. suecica* have been shown to exhibit anti-inflammatory activity against NO and IL-6 [22], while those from *Isochrysis galbana*, which is a strain rich in glucans, inhibited the proliferation of human leukemia cells [23]. Moreover, the sulfated polysaccharides from the genus *Porphyridium* spp., have been shown to prevent the accumulation and action of free radicals (ROS), acting protectively against oxidative stress [20,21].

The recovered E4 eluates of *T*. *verrucosa* f. *rubens* PLA1-2 and *T*. *suecica* T3-1 were evaluated for putative biological properties such as effects on the viability of human cells, on wound-healing, and on viral replication. Based on our results, there were no negative effects on the viability of primary or of cancer human cells, by the use of the anionic polysaccharides-containing fractions of the two *Tetraselmis* strains, irrespectively of their content of sulfated polysaccharides. In several studies, high and dose-dependent cytotoxicity has been observed when cancerous cell lines were treated with polysaccharides’ extracts obtained from microalgae (e.g., *Crypthecodinium cohnii*, *Galdieria sulphuraria*, *Isochrysis galbana*, *Nannochloropsis oculate*, *Porphyridium cruentum*, *Dixoniella grisea*, *Dunaliella salina*, etc.). Therefore, the microalgae-derived polysaccharides’ extracts are considered natural compounds bearing anti-cancer properties based on their effects on many human cancer cell lines; indicatively, MCF-7 (breast adenocarcinoma), HeLa (cervical carcinoma), HCT 116 (colorectal carcinoma), U-937 and HL-60 (leukemia), NCI-H460 (lung carcinoma), and HepG2 (hepatocellular liver carcinoma) [44,45,46,47,48]. Interestingly, studies involving non-cancerous cell lines have reported a lack of cytotoxic effects in cells treated with microalgal-derived polysaccharides. For example, no effects on cell viability were observed when Vero cells were treated with *Cladophora* spp. derived sulfated polysaccharides up to a concentration of 625 μg per mL of cell culture medium [49]. Moreover, fractions of polysaccharides from the red microalgae *Porphyridium cruentum* were shown to exhibit cytotoxic effects against several cancerous cell lines (colon carcinoma HT-29 cells, breast adenocarcinoma MCF-7 cells and triple negative mammary adenocarcinoma MDA-MB-231 cells), but did not reduce the viability of a non-cancerous cell line (mammary epithelial MCF-10A cells), when the same concentrations of polysaccharides were used to supplement the cells’ culture medium [50].

The absence of cytotoxic effects of E_4_ eluates of both *Tetraselmis* strains on the normal human dermal fibroblast cells, gave us the opportunity to use an in vitro assay, that simulates the wound healing procedure by examining the effects on cell migration, to investigate the putative wound healing properties of these fractions. The results showed that the E_4_ eluate from *Τ. suecica* T3-1 did not affect the migration of the normal human dermal fibroblast cells neither positively nor negatively, at any concentration applied. Interestingly, as shown in Table 3, the E_4_ eluate from *Τ. suecica* T3-1 had a low content of sulfated polysaccharides and did not exhibit antioxidant capacity. In contrast, a significant positive effect was observed when the E_4_ eluate from *T. verrucosa* f. *rubens* PLA1-2 was used, at both concentrations tested and this effect was also dose-dependent. The E_4_ eluate from *T. verrucosa* f. *rubens* PLA1-2 was characterized by notable antioxidant capacity and concentration of sulfated polysaccharides, thus these characteristics seemed to increase the migration of cells towards the scratched area. In vivo, studies should undoubtedly prove the efficacy of the specific eluate on wound healing.

Antioxidant, anti-inflammatory, antimicrobial, antiviral, and hygroscopic effects of sulfated polysaccharides are considered mandatory in skin regeneration [51,52]. By comparing the cell-migration effects of E4 eluates from the two microalgae strains, we conclude that the presence of sulfated polysaccharides bearing antioxidant capacity is crucial for enhancing the migration of the dermal fibroblast cells to restore the gap in the cell layer. Thus, our results are in agreement with numerous studies [53] showing the significant role of sulfated polysaccharides in wound treatment approaches. Indeed, microalgal (especially species belonging to *Chlorophyceae*, *Eustigmatophyceae*, *Chlorodendrophyceae*, and *Porphyridium* spp.) and cyanobacterial biologically active compounds, such as pigments, peptides, fatty acids, and polysaccharides have been widely used in the production of wound dressings for skin regeneration applications. Indicatively, extracts from *Spirulina* spp., *Aphanothece* spp., *Chlorella* spp., *Chlamydomonas* spp., *Tetraselmis* spp., and *Porphyridium* spp. have been examined for various potential effects (including biocompatibility, proliferation and migration of fibroblast cells, antithrombogenicity, antitumor effects against UVB irradiation, anti-photoaging effect, wound angiogenesis, and collagen deposition increase, IgE, TNF-α, IL-4, IL-5, INF-γ reduction, antimicrobial effects, etc.) in aqueous, ethanolic, or methanolic solutions, in simple, synthetic, or hydrogel form [26].

Extracellular polysaccharides isolated from *Nostoc* spp. PCC7936 strain, when cultivated under acetate ions supplementation in the medium were used for hydrogel formation in the presence of 0.4% (*w*/*v*) FeCl_3_. These biocompatible hydrogels promoted fibroblast migration and wound healing of scratches in NHDF [54]. Water-soluble extracts derived from *Chlorella vulgaris* and *Arthrospira platensis*, containing polysaccharides and proteins, and microencapsulated with sericin, presented cytocompatibility and significant human fibroblast cell migration and complete wound closure, especially when the extract from *A. platensis* was used [55]. In another study, Melo et al. [56] used extracts from *Chlorella vulgaris* cultivated auto- and mixotrophically. Hydrogels were prepared using Carbopol and tested for healing dermal wounds in Swiss albino mice. The hydrogel based on the extract from the mixotrophic cultivation resulted in an accelerated healing process.

In the current study, the E_4_ eluates from *T. verrucosa* f. *rubens* PLA1-2 and *Τ. suecica* T3-1 presented also significant antiviral properties against the Hepatitis C Virus (HCV). HCV has a positive sense single stranded RNA (+ssRNA) as a genome, and it is a blood-borne virus that infects the human liver causing acute or chronic hepatitis, which may progress to liver cirrhosis and hepatocellular carcinoma. According to Word Health Organization (WHO), about 50 million people are infected by HCV worldwide [57]. The viral RNA-dependent RNA-polymerase is error prone, thus, the virus circulates in the blood of the patient in the form of quasispecies making it difficult to develop a vaccine and increasing the likelihood of failure of the direct-acting antivirals, currently used as antiviral therapy, due to the emergence of new mutations. Thus, new agents with multi-level modes of action against HCV are needed.

The putative antiviral activity is probably the most studied property of microalgae-derived sulfated polysaccharides and these compounds have been shown to act as antiviral agents against many enveloped and non-enveloped viruses [43,58]. The proposed mechanisms of action of sulfated polysaccharides include possible changes on the surface of the virion or of the cell, and/or changes on the cell gene expression and metabolism, that could inhibit the virion attachment/penetration and uncoating, and/or suppress the expression and replication of the virus genome, respectively. As the E_4_ eluates from both *Tetraselmis* strains reduced the HCV replication, we conclude that this effect is irrespective of the content of sulfated polysaccharides or the presence of antioxidant capacity in these fractions. Extracts with a high degree of sulfation present anionic properties and usually act as protective agents against viruses [33]. Especially, extracellular polysaccharides derived from strains belonging to *Arthrospira* spp., *Porphyridium* spp., *Gyrodinium* spp., *Rhodella* spp., and *Cochlodinium* spp. have exhibited antiviral activity against viruses indicatively belonging in *Herpesviridae*, *Poxviridae*, *Paramyxoviridae*, *Orthomyxoviridae*, *Retroviridae*, *Hepadnaviridae*, etc. [13].

## 4. Materials and Methods

### 4.1. Microalgae Strains and Cultivation Conditions

Fourteen microalgae strains previously isolated from Greek coastal lagoons and deposited in the AthU-Al (Athens University Algae) strain bank (http://phycotheca.biol.uoa.gr) of the NKUA (National and Kapodistrian University of Athens, Greece), were used in this study. These strains included *Tetraselmis verrucosa* f. *rubens* PLA1-2, *T*. *verrucosa* f. *rubens* KSI1-3, *T*. *verrucosa* f. *rubens* KLE1-3, *T*. *verrucosa* f. *rubens* KLE1-4, *T*. *suecica* ELO1-1, *T*. *suecica* ELO1-2, *T*. *suecica* T3-1, *T*. spp. Mes5, *T*. spp. Mes17, *T*. spp. EST1-2, *Dunaliela salina* 30, *D*. *salina* 31, *D*. *salina* 32, and *Nannochloropsis gaditana* PLA1-1. The above microalgae were cultivated for 10 days in rigorously aerated 2-L flasks, using Walne’s solution of adjusted salinity 40‰ as a growth medium. The cultivated microalgae were incubated at 21–23 °C, air supplied through a pump at a flow rate of 0.5–1 vvm, at ~35% relative humidity, using a light period of 12 h day/12 h night and a light intensity of 100–150 μmoles × photons × m^−2^ × s^−1^ [59]. To obtain the dry biomass (DB), the grown microalgae were centrifuged at 5000× *g* for 10 min and the microalgae containing pellet was freeze-dried at −45 °C (VirTis Freezemobile 12SL, The Virtis Company, New York, NY, USA) to constant weight. Three independent cultivations and harvests were performed for each microalgae strain.

### 4.2. Determination of Total Sugars and β-Glucan Content of Microalgae

The total sugar content of microalgae was determined as described by Ntzouvaras et al. [59]. Briefly, the microalgae DB was suspended in ultra-pure water (ddH_2_O), 1 mL of the resulting suspension was mixed with 6 mL of phenol-sulfuric acid, incubated at 20 °C with continuous vortexing, and then, the absorbance was measured at 490 nm. A standard curve generated by the absorbance of known amounts of glucose was used to determine the total sugar content of microalgae as glucose equivalents. The results were expressed as g of sugars/100 g of microalgae DB (%, *w*/*w*).

The determination of *β*-glucans content (g/100 g DB) was conducted using the *β*-glucan enzyme assay kit (K-YBGL) of Megazyme^®^ (Wicklow, Ireland), following the manufacturer’s instructions. Only microalgae that exhibited a total sugar content above 20% *w*/*w* were checked with this assay. The *β*-glucan enzyme assay determines directly the total and *α*-glucans and indirectly the *β*-glucans, and it has been previously used to quantify the *β*-glucans of microalgae [38,60,61,62]. Briefly, the quantification of total glucans involved hydrolysis of the microalgae DB in the presence of concentrated sulfuric acid (H_2_SO_4_, acidic hydrolysis), followed by heating at 100 °C for 2 h to release the monosaccharides. After pH adjustment, the remaining glucan fragments were quantitatively hydrolyzed to glucose molecules using a mixture of highly purified exo-1,3-*β*-glucanase and *β*-glucosidase enzymes. Then, the released glucose molecules were incubated with glucose oxidase and peroxidase in the presence of 4-aminoantipyrine and p-hydroxybenzoic acid, and the absorbance of the resulting product was measured at 510 nm against the reagent blank, using a Hitachi U-2000 spectrophotometer (Hitachi Ltd., Tokyo, Japan). Known concentrations of glucose were treated as above (with glucose oxidase and peroxidase) and the measured values of absorbance at 510 nm were used to create a standard curve. The total glucan content (% *w*/*w*) of each microalgae sample was estimated based on the following type:Total glucan content (% *w*/*w*) = ΔA × F/W × 90(1)
where: ΔA = sample reaction absorbance − reagent blank absorbance, F = absorbance value of the D-glucose 100 μg standard, W = weight of sample analyzed.

For the quantification of *α*-glucans, the microalgae DB was hydrolyzed in the presence of concentrated sodium hydroxide (NaOH, alkaline hydrolysis), followed by adjustment of pH and enzymatic treatment with amyloglucosidase and invertase. The released glucose molecules were treated and quantified as described above. The *α*-glucan content (% *w*/*w*) of each microalgae sample was estimated based on the following type: *α*-glucan (% *w*/*w*) = ΔA × F/W × 9.315

The *β*-glucan content of each microalgae sample was calculated indirectly as the difference between total glucan content and *α*-glucan content. In each process, three independent DB samples were analyzed per microalgae strain.

### 4.3. Extraction and Isolation of Crude Polysaccharides from Microalgae

The isolation of crude polysaccharides of microalgae involves several steps: (1) the removal of lipophilic molecules from the microalgae DB, (2) the disruption of microalgae to release the water-soluble intracellular, structural, and bound molecules, (3) the deproteinization process, and finally, (4) the recovery of extracts containing the crude polysaccharides.

#### 4.3.1. Extraction of Lipophilic Molecules from the Microalgae DB

Lipophilic molecules (lipids, chlorophyll, and other pigments) are extracted in organic solvents [63]. Thus, the microalgal DB was suspended in acetone under continuous vortexing and centrifuged at 5000× *g* for 10 min at 4 °C (Hettich Universal Centrifuge, Model 320-R, Merck KGaA, Darmstadt, Germany). The resulting pellet was suspended in acetone and treated again as above. The procedure was repeated several times until the complete decolorization of the microalgal biomass. Residual acetone in the defatted microalgal biomass was removed by evaporation, after drying the biomass at 50 °C under vacuum, using a Flash Evaporator/Rotavapor R-114 system (BÜCHI Labortechnik AG, St. Gallen, Switzerland), for several rounds of 30 min (until biomass weight stabilization).

#### 4.3.2. Microalgae Cells’ Disruption Process

To facilitate microalgae cell disruption and molecule extraction, the defatted microalgal biomass was suspended in ddH_2_O, and ultrasounds, thermal treatment (incubation in hot water—100 °C), and vortexing techniques were used, according to a slightly modified protocol based on Kurd and Samavati [64] and Huo et al. [65]. The high-frequency ultrasounds were produced by an Ultraschallprozessor Hielscher UP200S device (Hielscher Ultrasonics, Teltow, Germany) (power: 200 W), set at 80% of its power. Ultrasound treatment of each sample lasted 10–15 min with intermediate pauses, while the temperature remained between 80–100 °C, due to the ultrasounds. Then, each sample was centrifuged at 5000× *g* for 10 min. The supernatant containing the crude water-soluble fraction was kept for further treatment, whereas the pellet was treated once again as above to maximize the extraction of molecules. After centrifugation, the new supernatant was kept and mixed with the previous one and further processed.

#### 4.3.3. Deproteinization Process of the Crude Water-Soluble Fraction of the Defatted Microalgal Biomass

The deproteinization procedure was carried out using a 5:1 chloroform/n-butanol solvent mixture (Sevag method) to insolubilize and remove the proteins from the extract through centrifugation. Briefly, the method involved the mixing of the crude water-soluble fraction of the defatted microalgal biomass and the solvent mixture at a 2:1 ratio, followed by vigorous vortexing and centrifugation at 15,000× *g* for 10 min at 4 °C. This resulted in the formation of two phases (an aqueous upper phase and an organic lower phase) and an interface containing the denatured protein precipitate. The interface and the organic phase were removed, while the aqueous phase was mixed with solvent mixture at a 2:1 ratio, vortexed, and centrifuged, and the procedure was repeated until the interface was no longer visible. At least six repetitions were required for the specific extracts, as recommended by Huo et al. [65].

The removal of proteins was confirmed by using the Bradford assay (with detection limits of 0.1–1.0 mg/mL) [66]. Specifically, for each deproteinated sample a small volume (100 µL) was mixed with 5 mL of Bradford reagent 1X (Sigma-Aldrich, St. Louis, MO, USA), vortexed, incubated for 5 min, and then its absorbance was measured at 595 nm. A standard curve generated by the absorbance at 595 nm of known amounts of bovine serum albumin (BSA) protein (Panreac, Barcelona, Spain) was produced and used to quantify the total protein content of the sample as albumin equivalents (g/L). All samples were analyzed in triplicates.

#### 4.3.4. Isolation of Crude Polysaccharides Extract from the Deproteinated Samples

To isolate the polysaccharides, the deproteinated sample was mixed with cold absolute ethanol (96% *v*/*v* purity) in a 1:4 ratio. The mixture was vortexed for 1 h, incubated for 24 h at 4 °C, and then centrifuged at 15,000× *g* for 10 min at 4 °C, to precipitate the polysaccharides. The pellet of polysaccharides was dissolved in ddH_2_O and frozen.

### 4.4. Detection and Quantification of Sugar Monomers of the Polysaccharide Extracts

The polysaccharide extracts were further analyzed to detect the quantity of their respective sugar monomers. For each sample (10.0 mg/mL concentration), 0.5 mL underwent acidic hydrolysis using 1 mL of trifluoroacetic acid (TFA) 4 Μ, combined with heat treatment (100 °C for 3 h), to release sugar monomers. The resulting hydrolysates were neutralized using an equal volume of NaOH 4 Μ. Identification and quantification of monosaccharides were carried out using a modified protocol based on 1-phenyl-3-methyl-5-pyrazolone (PMP)-derivatization process adapted from Dai et al. [67] and Ai et al. [68]. Briefly, 0.5 mL of the hydrolyzed sample was mixed with 0.8 mL of 0.5 M PMP and 1.2 mL of 0.3 M NaOH and heated in a water bath at 70 °C for 1 h, to facilitate PMP-derivatives. The sample then was cooled using a cold-water bath and 1.2 mL of 0.3 M HCl was added to neutralize the pH and terminate the reaction. To remove excess PMP, 1 mL of chloroform was added to the mixture, followed by vortexing and centrifugation at 17,000× *g* for 10 min, resulting in the formation of two phases. The lower phase (organic) that contained the PMP was removed. For the complete removal of PMP the above process was repeated 2–3 times and then, the aqueous phase was subjected to High Performance Liquid Chromatography (HPLC) using a Waters Alliance 2695 apparatus (Waters Corporation, Milford, MA, USA) equipped with a RP-HPLC Column SVEA C18 Plus (4.6 mm × 250.0 mm, 5.0 μm) (Thermo Fisher Scientific, Waltham, MA, USA) and a Waters 2696 PhotoDAD (Diode Array Detector) (Waters Corporation, Milford, MA, USA) for UV detection at 245 nm. The injection volume was 20 μL with an eluant flow rate of 0.8 mL/min. Two mobile phases were used, and gradient elution was performed. Mobile phase A was 100% acetonitrile and phase B was a mixture of ddH_2_O and acetonitrile (9:1, *v*/*v*) with 0.045% KH_2_PO_4_ and 0.05% triethylamine buffer (pH 7.5), as described by Ai et al. [68]. The identification of the sugar monomers was based on the retention time of each substance, compared to the retention time of standards (i.e., arabinose, fucose, galactose, glucose, mannose, rhamnose, ribose, glucuronic acid, galacturonic acid, and xylose). The peak area ratio of the analyte relative to that of the respective standard and the concentration of the standard were used to calculate the content (%, *w*/*w*) of each sugar monomer in the polysaccharide extract. Three independent samples were analyzed per microalgae strain.

### 4.5. Fractionation of Polysaccharide Extracts and Eluates Characterization

For each polysaccharide extract, fractionation was performed via ion-exchange chromatography, using as an anion exchanger the diethylaminoethyl cellulose resin (DEAE Cellulose 52) (Sigma-Aldrich, St. Louis, MO, USA), to retain the negatively charged molecules. Chromatography was performed using a 15.0 mm × 100 mm Glass Fritted Chromatography Column (Bio-Rad, Hercules, CA, USA) with 5 mL of stationary phase. The resin layer was washed with 20 mL of ddH_2_O, then 1 mL of the polysaccharide extract was added to the column, and the eluate was recovered (E1). Then, 1 mL pure water (ddH_2_O) was loaded onto the column and the eluate (E2) was obtained. Next, the eluates E3 and E4 were obtained after loading the column with NaCl solutions using a gradually increasing NaCl concentration (0.2 and 1.0 M, respectively), to elute the negatively charged molecules bound to the column. The NaCl was removed from the eluates through filtration using Amicon Ultra-2 centrifugal filters (with cut-off 3 kDa) (Sigma-Aldrich, St. Louis, MO, USA), according to the manufacturer’s instructions. The eluates were analyzed to determine the total sugars as described above (Materials and Methods Section 4.2), and further characterized concerning the concentration of sulfates and their antioxidant activity (see below).

#### 4.5.1. Quantification of Eluates’ Sulfates of the Polysaccharide Extracts

The determination of sulfates (SO_4_^−^, mg/mL) in the eluates derived from the fractionation of the polysaccharide extracts, was carried out based on the protocol developed by Torres et al. [69]. Briefly, 20 µL of appropriately diluted eluate (detection limits of sulfates: 0.01–1.22 mg/mL) was mixed with 140 µL of 0.5 M HCl and 40 µL of barium chloride-gelatin reagent, in a flat-bottom 96-well clear plate (SPL Lifescience, Pocheon, Republic of Korea). After 30 min of incubation, the absorbance was measured at 405 nm. Three independent samples were analyzed per microalgae strain. The quantification of sulfates (mg/mL) was based on a standard curve generated by the absorbance at 405 nm of known concentrations of SO_4_^−^, derived from Na_2_SO_4_ (0.676 mg SO_4_^−^ per mg Na_2_SO_4_). To produce the barium chloride-gelatin reagent, 25 mg of gelatin (Sigma-Aldrich, St. Louis, MO, USA) was mixed with 25 mL of ddH_2_O and incubated for 10 min under gentle agitation at 80 °C. Subsequently, 250 mg of BaCl_2_ (Sigma-Aldrich, St. Louis, MO, USA) was added and stirred to homogenize.

#### 4.5.2. Antioxidant Capacity of the Eluates of the Polysaccharide Extracts

The antioxidant capacity of the eluates of the polysaccharide extracts was evaluated using the Ferric Reducing Antioxidant Power (FRAP) assay, by monitoring the capacity of the extract to reduce ferric tripyridyl triazine to ferrous triazine, as previously described by Wang et al. [70], with minor modifications. Briefly, 10 µL of appropriately diluted eluate (detection limits: 20–200 µM Trolox equivalents, see below) was mixed with 190 µL of FRAP reagent, vortexed, and incubated at 37 °C for 30 min. Absorbance was measured at 593 nm using a microplate reader (Tecan, Männedorf, Switzerland). A standard curve of the FRAP assay was generated using known amounts of the 6-hydroxy-2,5,7,8-tetramethylchroman-2-carboxylic acid (Trolox) (Sigma-Aldrich, St. Louis, MO, USA) and used to quantify the total antioxidant capacity of the eluates as Trolox equivalents (µM). Three independent samples were analyzed per microalgae strain.

#### 4.5.3. Attenuated Total Reflection—Fourier Transform Infrared (ATR-FTIR) Analysis

The ATR-FTIR spectra of the *T. verrucosa* f. *rubens* PLA1-2 polysaccharide extract (E4 eluate) were obtained by a Perkin Elmer Spectrum-Two spectrometer equipped with a Diamond ATR compartment (Perkin Elmer, Hopkinton, MA, USA) using the Spectrum 10 software (v.10.5.1.581) provided by the manufacturer. The recorded spectrum was an average of 64 scans in the mid-infrared region (4000 and 400 cm^−1^) at a resolution of 4 cm^−1^. The recorded spectrum was then ATR-corrected using a refractive index of 1.5 for diamond, to be comparable to the available spectral libraries for facilitating the interpretation of spectra, linear baseline was corrected and then normalized by the mean using The Unscrambler X v.10.5 software (CAMO software, Oslo, Norway).

### 4.6. Biological Properties of the Eluates of the Polysaccharide Extracts

Selected eluates (E1-4) of the polysaccharide extracts were subsequently assessed for effects on the viability of human cells and for putative antiviral and wound healing properties. To do so the polysaccharide extract eluates were freeze-dried and then a specific amount was dissolved in ddH_2_O.

#### 4.6.1. Effects on the Viability of Human Cells

Putative effects of the polysaccharide extract eluates on human cell viability were assessed in vitro using normal human dermal fibroblasts (NHDF-Ad—Human Dermal Fibroblasts, Adult) (Lonza Group, Brussels, Switzerland) cells (primary cells), whereas the cell viability was evaluated with the 3-[4,5-dimethylthiazol-2-yl]-2,5 diphenyl tetrazolium bromide (MTT) assay. This assay is based on the conversion of the yellow MTT into purple formazan crystals, by the mitochondrial activity of the living cells. Formazan crystals are dissolved in DMSO and the absorbance at 560 nm of the resulting solution is used to estimate the viability of cells.

Prior to the assay, NHDF cells were routinely cultivated in a Fibroblast Growth Medium-2 (FGM-2 Bulletkit, Lonza Group, Brussels, Switzerland) at 37 °C and 5% CO_2_ and subcultured with ReagentPack™ Subculture kit (Lonza Group, Brussels, Switzerland), following the instructions of the manufacturer. To check for effects of a polysaccharide extract eluate on cell viability, NHDF cells were seeded in 96-wells sterile culture plates (Nunc™, Thermo Fisher Scientific, Rockford, IL, USA) (10,000 cells/well) and cultivated in fibroblast growth media-2 (FGM-2 medium; Lonza, Walkersville, MD, USA) for 24 h. Then, the culture medium was replaced with a new one containing different amounts of the polysaccharide extract eluate (concentrations used: 50–500 μg/mL) or the equal volume of ddH_2_O (control), and the cells were further incubated for 24 and 48 h. Subsequently, 20 μL of MTT (Sigma-Aldrich, St. Louis, MO, USA) (5 mg/mL) was added to each well, the cells were incubated for 2 h, then 200 μL dimethylsulfoxide (DMSO; Panreac, Barcelona, Spain) was added, to dissolve the formed purple formazan crystals, and the absorbance was measured at 560 nm. Background control wells contained only the culture medium and MTT reagent, without cells, to account for any non-specific absorbance and ensure accurate measurement of cell viability. All samples were analyzed in quadruplicate and the cell viability was calculated according to the following equation:Cell Viability due to tested sample (%) = (Absorbance of the test at 560 nm)/(Absorbance of the control at 560 nm) ×100)

#### 4.6.2. Wound Healing Effects

For the assessment of the putative wound healing properties of the polysaccharide extract eluates, approximately 20,000 NHDF cells per well were seeded in the Incucyte^®^ Imagelock 96-Well Plate (Essen BioScience, Sartorius, Royston, UK) and incubated for 24 h. Then, standardized wounds were created using Wound Maker™ (Essen BioScience, Sartorius, Royston, UK). The culture medium was carefully removed, the cells were washed with PBS 1X and a new culture medium that contained a lower amount of fetal calf serum (FCS) (1%) to reduce cell proliferation, and was supplemented with a polysaccharide extract eluate, was added. Two different concentrations (300 and 500 μg/mL) of polysaccharide extract eluates were used, whereas the cells used as controls in the respective new culture medium were supplemented with an equal volume of ddH_2_O. Then, the cells were incubated at 37 °C, 5% (*v*/*v*) CO_2_ in a humidified incubator connected with IncuCyte S3 apparatus (Essen BioScience, Sartorius, Royston, UK) to monitor cell migration (magnification ×10, photo shooting every three hours for 48 h). After this period, the results of each time point were collected and analyzed, and the relative healing of the wound was determined using the Incucyte^®^ ZOOM™ software (v2009B) (Essen BioScience, Sartorius, Royston, UK).

#### 4.6.3. Antiviral Activity Properties

The putative antiviral activity of the polysaccharide extract eluates was examined in the human hepatoma Huh5-2 cell line stably harboring the subgenomic reporter replicon of the Hepatitis C virus (HCV) genotype 1b (Con1 strain) [31]. This HCV subgenomic reporter replicon autonomously replicates in the Huh5-2 cells and expresses a selectable marker (the neo gene), which encodes the enzyme aminoglycoside 3′-phosphotransferase conferring resistance to geneticin (G418), and the reporter protein firefly luciferase (F-Luc). The level of F-Luc activity directly correlates with the level of the replication of this HCV replicon [31]. The Huh5-2 cells harboring the HCV reporter replicon were routinely cultured using high glucose (25 mM) Dulbecco’s modified minimal essential medium (Thermo Fisher Scientific, Waltham, MA, USA), supplemented with 0.1 mM non-essential amino acids, 2 mM L-glutamine, 100 U/mL penicillin, 100 µg/mL streptomycin, 10% (*v*/*v*) heat inactivated FCS, and 500 µg/mL geneticin (G418) [71] and incubated at 37 °C, 5% (*v*/*v*) CO_2_ in a humidified incubator.

To check for the effects of a polysaccharide extract eluate on the replication of the HCV, the cells were seeded in 96-wells plates. At 24 h the culture medium was replaced with a new one which did not contain geneticin but contained different amounts of the polysaccharide extract eluate (concentrations used: 50, 100, or 200 μg/mL) or the equal volume of ddH_2_O (control). The cells were further incubated for 72 h and then, the cells were washed twice with phosphate buffered saline (PBS) 1X and lysed in 40 μL of Luciferase Cell Culture Lysis Reagent 1X (Promega Corporation, Madison, WI, USA). F-Luc activity was determined in cell lysates, using the Luciferase Assay system (Promega Corporation, Madison, WI, USA) according to the manufacturer’s instructions. Briefly, 4 μL of the cell lysate was mixed with 30 μL of Luciferase Assay Reagent, vortexed immediately and the produced bioluminescence was measured using a GloMax 20/20 single tube luminometer (Promega Corporation, Madison, WI, USA) for 10 s. Luciferase activity was normalized to the total protein amount quantified with the Bradford assay and expressed as relative light units per μg of protein (RLU/μg). The median effective concentration (EC_50_) of the polysaccharide extract eluate, refers to the concentration that reduces the F-Luc signal by 50% compared to the control and it was determined by nonlinear regression analysis using the Prism 6.0 software (GraphPad Software Inc., La Jolla, CA, USA).

## 5. Conclusions

Microalgae have been demonstrated to be valuable sources of molecules, including bioactive polysaccharides. The microalgae strains studied here exhibited heterogeneity regarding the total sugar content. Among them, *Τ*. *suecica* T3-1 and *T*. *verrucosa* f. *rubens* PLA1-2 contained significant amounts of *β*-glucans, followed by galactans. The anionic eluates of both microalgae showed no cytotoxicity in normal (NHDF) and cancer (Huh5-2) cells even in high concentrations but displayed significant antiviral activity against the Hepatitis C virus replicon. The anionic eluate of *T*. *verrucosa* f. *rubens* PLA1-2 contained notable sulfates, presented antioxidant capacity, and was found to increase the migration of the NHDF cells in a wound healing simulating assay, thus it could be potentially utilized as a wound healing agent in skin regeneration.

## Figures and Tables

**Figure 1 marinedrugs-23-00077-f001:**
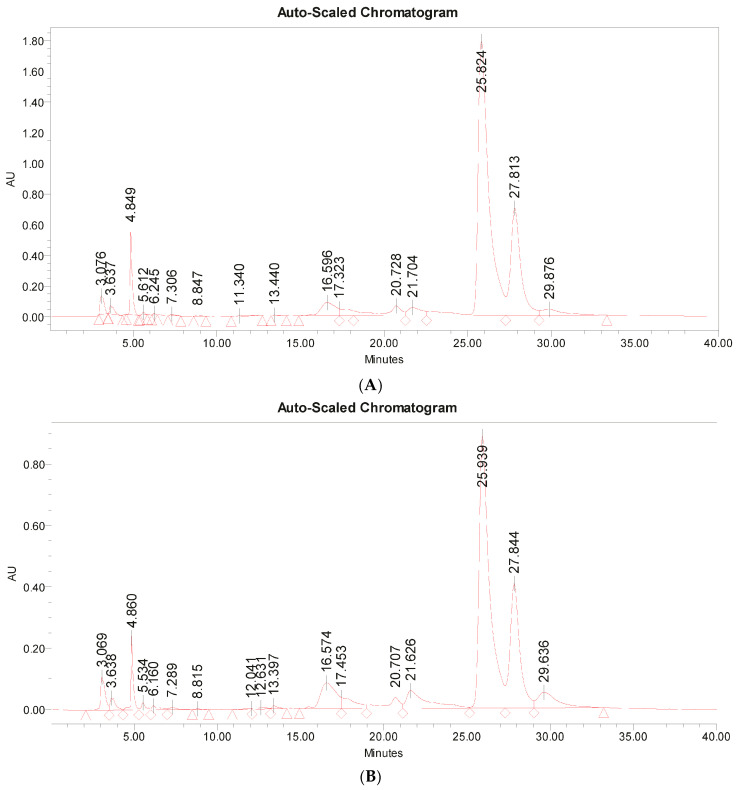
HPLC chromatogram of hydrolyzed polysaccharides. (**A**) HPLC chromatogram of hydrolyzed polysaccharides derived from *T*. *verrucosa* f. *rubens* PLA1-2. (**B**) HPLC chromatogram of hydrolyzed polysaccharides derived from *T*. *suecica* T3-1. Monosaccharides were identified based on the retention time of the respective standards as follows: mannose—16.5 min, ribose—17.4 min, rhamnose—20.7 min, glucuronic acid—21.6 min, galacturonic acid—23.3 min, glucose—25.8 min, galactose—27.8 min, xylose—29.6 min, arabinose—30.5 min, fucose—35.1 min. Based on the retention time of the sugar standards, the following sugars were identified in the above chromatograms of the hydrolyzed polysaccharides of the two microalgal strains: glucose, galactose, mannose, xylose, rhamnose, and ribose.

**Figure 2 marinedrugs-23-00077-f002:**
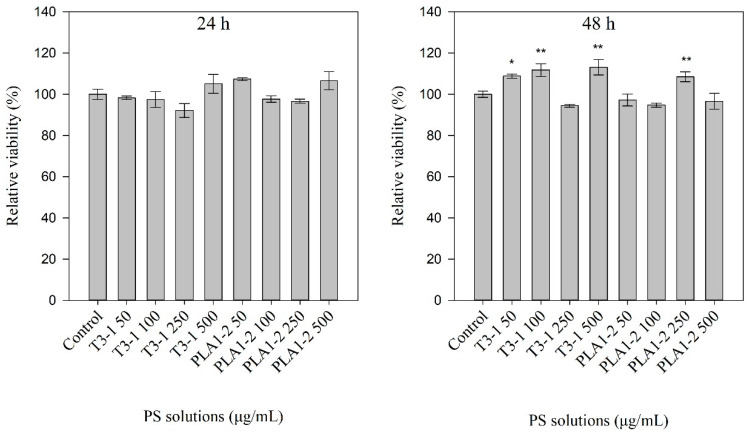
Effect of E4 eluates on cell viability. Relative viability of normal human dermal fibroblasts (NHDF) when incubated (for 24 h-left panel, for 48 h-right panel) in the presence of various concentrations of E_4_ eluates derived from the microalgae strains *Τ*. *suecica* T3-1 (T3-1) and *T*. *verrucosa* f. *rubens* PLA1-2 (PLA1-2). The different concentrations of 50, 100, 250, and 500 μg of anionic polysaccharides/mL of cell culture medium, used are mentioned. Control cells were incubated with culture medium supplemented with the respective volume of ddH_2_O (which was equal to the volume added in the case of the E_4_ eluates). The viability of the cells at the control assay was set to 100%. Bars represent mean values from three independent experiments in triplicate. Error bars indicate standard deviations. * *p* < 0.05, ** *p* < 0.01, versus control assay’s results.

**Figure 3 marinedrugs-23-00077-f003:**
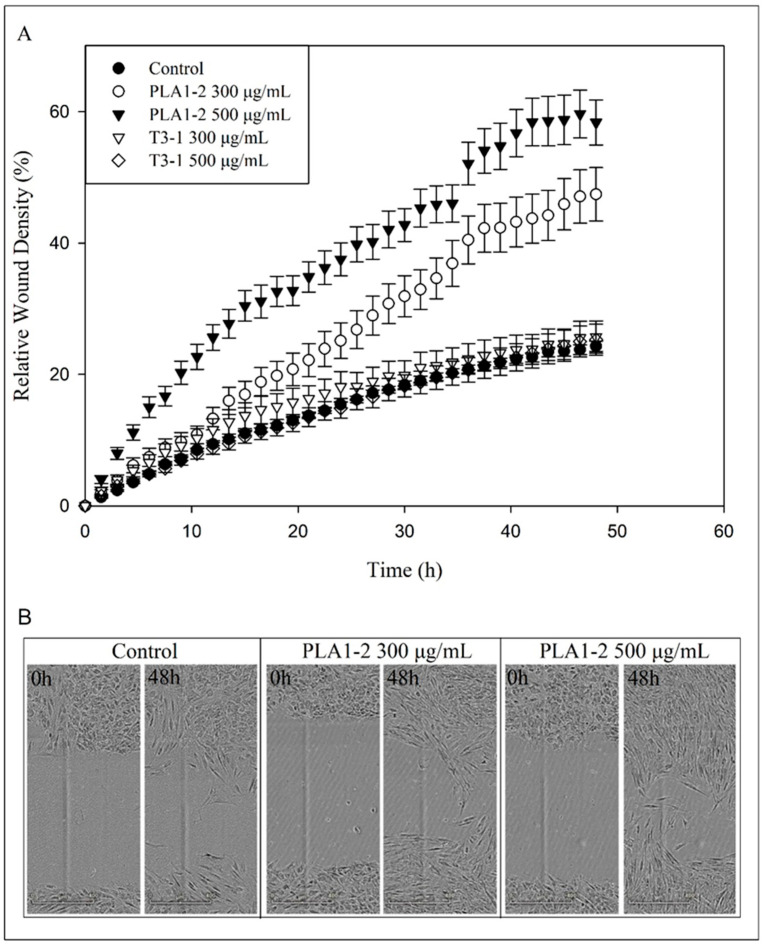
Effect of E4 eluates on wound healing. At t = 0 h a scratch was produced in the layer of NHDF cells, and the culture medium was changed with new one supplemented, or not (control cells), with 300 μg or 500 μg of the strongly anionic polysaccharides of a *Tetraselmis* strain per mL of cell culture medium, and the migration of cells, to close the scratch-produced gap in the cell layer, was monitored every 3 h for 48 h. (**A**) Quantification results of cell migration based on the images of the gap in the layer of cells, at 3 h-intervals post treatment and up to 48 h. Relative wound density achieved using E_4_ extracts from *T*. *verrucosa* f. *rubens* PLA1-2 and *Τ*. *suecica* T3-1 as healing agents. The relative wound density is the ratio of the cell occupied area to the area of the initial gap (set at 100%) in the layer of cells. Bars represent mean values from three independent experiments in triplicate. Error bars indicate standard deviations. (**B**) Representative images of the gap in the layer of cells at 0 h and 48 h post treatment. Control cells and cells treated with300 μg, and 500 μg, of the strongly anionic polysaccharides from *T*. *verrucosa* f. *rubens* PLA1-2. Control cells were incubated with culture medium supplemented with the respective volume of ddH_2_O (which was equal to the volume added in the case of the E_4_ eluates).

**Figure 4 marinedrugs-23-00077-f004:**
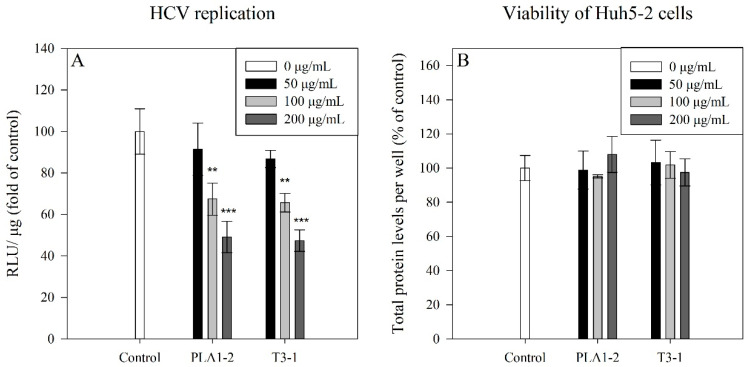
Effect of E4 eluates on HCV replication. Huh5-2 cells stably harboring the subgenomic reporter replicon of HCV that expresses the firefly luciferase (F-Luc), were treated for 72 h with different concentrations, 50 μg, 100 μg, or 200 μg of the strongly anionic polysaccharides of a *Tetraselmis* strain per mL of cell culture medium. Control cells were incubated with culture medium supplemented with the respective volume of ddH_2_O (which was equal to the volume added in the case of the E_4_ eluates). (**A**) Viral RNA replication-derived firefly luciferase activity in the presence of various concentrations of E_4_ eluates derived from the microalgae strains *Τ*. *suecica* T3-1 (T3-1) and *T*. *verrucosa* f. *rubens* PLA1-2 (PLA1-2). The different concentrations of 50, 100, and 200 μg of anionic polysaccharides per mL of cell culture medium, used are mentioned. The mean value of the firefly luciferase activity of control cells (expressed as relative light units (RLU)/μg of protein) was set to 100%. (**B**) Viability of cells in the presence of the anionic polysaccharides from the two *Tetraselmis* strains expressed as total protein levels per well. Bars represent mean values from three independent experiments in triplicate. Error bars indicate standard deviations. ** *p* < 0.01, *** *p* < 0.001, versus control.

**Table 1 marinedrugs-23-00077-t001:** Quantitative determination of total sugars in the dry biomass (DB) of 14 microalgae strains, *α*- and *β*-glucans, and *β*-glucans to total sugars ratio in selected strains.

	Strain	Total Sugars (g/100 g DB)	Total Glucans (g/100 g DB)	*α*-Glucans (g/100 g DB)	*β*-Glucans (g/100 g DB)	*β*-Glucans (g) Per 100 g of Total Sugars
1	*Tetraselmis verrucosa* f. *rubens* PLA1-2	40.9 ± 2.6	28.7 ± 2.0	17.7 ± 0.8	11.0 ± 1.0	26.9 ± 0.9
2	*T. suecica* T3-1	32.6 ± 1.4	20.8 ± 1.1	12.7 ± 0.4	8.1 ± 0.7	24.8 ± 1.1
3	*T. verrucosa* f. *rubens* KSI1-3	27.8 ± 0.9	12.6 ± 0.7	8.6 ± 0.2	4.0 ± 0.5	14.4 ± 1.3
4	*T. suecica* ELO1-1	20.2 ± 1.8	15.5 ± 1.0	12.6 ± 0.6	2.9 ± 0.4	14.4 ± 0.6
5	*Tetraselmis* spp. Mes5	20.1 ± 1.1	13.6 ± 0.9	12.6 ± 0.7	1.0 ± 0.1	5.0 ± 0.2
6	*Dunaliela salina* 32	19.8 ± 1.3	-	-	-	-
7	*Tetraselmis* spp. Mes17	18.5 ± 1.3	-	-	-	-
8	*T. suecica* ELO1-2	17.8 ± 1.9	-	-	-	-
9	*T. verrucosa* f. *rubens* KLE1-3	14.9 ± 0.7	-	-	-	-
10	*Nannochloropsis gaditana* PLA1-1	12.6 ± 1.9	-	-	-	-
11	*D. salina* 30	12.5 ± 0.7	-	-	-	-
12	*T. verrucosa* f. *rubens* KLE1-4	11.8 ± 0.4	-	-	-	-
13	*D. salina* 31	6.5 ± 0.8	-	-	-	-
14	*Tetraselmis* spp. EST1-2	5.5 ± 0.2	-	-	-	-

-: not determined; Data represent mean ± SD of three independent samples.

**Table 2 marinedrugs-23-00077-t002:** Results from the HPLC analysis of polysaccharide hydrolysates derived from the selected microalgae strains: identified sugar monomers, their percentage content (% or mg/100 mg of total sugars), and molar ratio (mmol/100 mg of total sugars).

Monomers	Strains			
	*T. verrucosa* f. *rubens* PLA1-2		*T. suecica* T3-1	
	(%)	(mmol/100 mg)	(%)	(mmol/100 mg)
Glucose	71.9 ± 3.1	0.40 ± 0.02	66.4 ± 1.1	0.37 ± 0.01
Galactose	19.8 ± 1.6	0.11 ± 0.01	20.9 ± 1.3	0.12 ± 0.00
Glucuronic acid	2.7 ± 0.1	0.01 ± 0.00	3.7 ± 0.4	0.02 ± 0.00
Mannose	2.1 ± 0.3	0.01 ± 0.00	3.7 ± 0.6	0.02 ± 0.00
Xylose	1.3 ± 0.2	0.01 ± 0.00	2.9 ± 0.6	0.02 ± 0.00
Ramnose	1.3 ± 0.1	0.01 ± 0.00	1.0 ± 0.0	0.01 ± 0.00
Ribose	0.8 ± 0.2	0.01 ± 0.00	1.4 ± 0.2	0.01 ± 0.00
Arabinose	-	-	-	-
Fucose	-	-	-	-
Galacturonic acid	-	-	-	-

-: not detected; Data represent mean ± SD of three independent samples.

**Table 3 marinedrugs-23-00077-t003:** Total sugars, sulfates, sulfates to total sugars ratio (mmol SO_4_^−^ per mg of total sugars), and antioxidant capacity (Trolox equivalents) of the fractions obtained by ion-exchange chromatography from the H_2_O-dissolved polysaccharides of the two *Tetraselmis* strains.

Strain	Eluates	Total Sugars (mg/mL)	SO_4_^− ^ (mg/mL)	SO_4_^−^/ Total Sugars (mmol/mg)	Antioxidant Capacity (μΜ Trolox Eq.)
*T. verrucosa* f. *rubens* PLA1-2	Ε_1_	0.68 ± 0.04	-	-	-
	Ε_2_	2.06 ± 0.35	-	-	-
	Ε_3_	0.66 ± 0.02	-	-	-
	Ε_4_	0.66 ± 0.06	0.485 ± 0.010	7.65 ± 0.93	3.00 ± 0.71
*T. suecica* T3-1	Ε_1_	0.59 ± 0.01	-	-	-
	Ε_2_	2.06 ± 0.07	-	-	-
	Ε_3_	1.09 ± 0.05	-	-	-
	Ε_4_	0.59 ± 0.07	0.014 ± 0.004	0.25 ± 0.11	-

-: not detected; Data represent mean ± SD of three independent samples.

## Data Availability

The original data presented in the study are included in the article/Appendix A; further inquiries can be directed to the corresponding author.

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
