# Peer review of "Wound Healing, Antioxidant, and Antiviral Properties of Bioactive Polysaccharides of Microalgae Strains Isolated from Greek Coastal Lagoons"

_marinedrugs, 2025, doi:10.3390/md23020077_

Round 1
Reviewer 1 Report
Comments and Suggestions for Authors
The study investigated 14 newly isolated microalgae strains from Greek coastal lagoons, focusing on their polysaccharide content and biological properties. The research revealed significant heterogeneity in total sugar content (5.5-40.9% of dry biomass) among strains, with Tetraselmis verrucosa f. rubens PLA1-2 and T. suecica T3-1 showing notably high β-glucan content.
The isolated polysaccharides were fractionated using ion-exchange chromatography, with particular attention to their anionic fractions. The anionic polysaccharides demonstrated no cytotoxicity against normal human cells while exhibiting some antiviral activity against Hepatitis C Virus. Notably, T. verrucosa f. rubens PLA1-2's anionic fraction contained sulfate groups, showed antioxidant capacity, and wound healing properties on the scratch cells assay.
The authors employed the following methods to characterized the polysaccharide:
- Chemical hydrolysis with TFA followed by HPLC analysis for monosaccharide composition
- PMP-derivatization process for sugar monomer identification
- Ion-exchange chromatography for sulfated polysaccharide separation
- Quantification of sulfate content in different eluates
Limitations and Suggested Improvements:
1. Spectroscopic Analysis Needed:
- FTIR would provide valuable information about functional groups and sulfate positioning
- NMR analysis would help determine:
* Detailed structural features
* Linkage patterns between monosaccharides
* Sulfation positions on the polymer chain
2. Data Presentation:
- Current table format should be modified to express hexose and sulfate content in molar ratios rather than mass percentages
- This would provide better understanding of the sulfation degree per sugar unit
3. Although the scratch cells assay is highly used to investigate would-healing effect, there are several limitations and should be accompained by additional methods.
- Being an in vitro assay, it does not fully replicate the complex environment of living tissues, which includes interactions with other cell types, extracellular matrix components, and signaling molecules.
- Lack of Mechanical and Biochemical Complexity: The assay does not account for the mechanical forces and biochemical gradients present in actual wound healing.
-
Cell Type Dependency: The results can vary significantly depending on the cell type used, which may not accurately represent the behavior of other cell types involved in wound healing in vivo.
These additional analyses would significantly strengthen the structural characterization of the polysaccharides and provide more comprehensive structural information.
Author Response
Dear Reviewer,
We would like to thank you for your comments, which we will attempt to address promptly.
Limitations and Suggested Improvements:
- Spectroscopic Analysis Needed:
- FTIR would provide valuable information about functional groups and sulfate positioning
- NMR analysis would help determine:
* Detailed structural features
* Linkage patterns between monosaccharides
* Sulfation positions on the polymer chain
Response: Concerning the suggestion to use FTIR to further characterize the functional sulfate groups positioning on the anionic polysaccharides of T. verrucosa f. rubens PLA1-2 eluate E4, we had intentionally avoided this because we believed that such a detailed physicochemical characterization fell outside the scope of the study. The aim was to identify microalgal strains and eluates with bioactive properties, with the intention of addressing the physicochemical characterization of these polysaccharides in a subsequent publication. However, we proceeded with an FTIR analysis of this specific eluate at this manuscript (see ‘Materials & Methods’ – L672-681), as recommended. The results of the analysis and the spectrum are presented at Results Section – Subsection ‘2.3.’ (L179-198) and at supplemental Figure S1, respectively.
- Data Presentation:
- Current table format should be modified to express hexose and sulfate content in molar ratios rather than mass percentages
- This would provide better understanding of the sulfation degree per sugar unit
Response: Thank you very much for your recommendation; Tables 2 and 3 have been modified accordingly.
- Although the scratch cells assay is highly used to investigate wound-healing effect, there are several limitations and should be accompanied by additional methods.
- Being an in vitro assay, it does not fully replicate the complex environment of living tissues, which includes interactions with other cell types, extracellular matrix components, and signaling molecules.
- Lack of Mechanical and Biochemical Complexity: The assay does not account for the mechanical forces and biochemical gradients present in actual wound healing.
-Cell Type Dependency: The results can vary significantly depending on the cell type used, which may not accurately represent the behavior of other cell types involved in wound healing in vivo.
These additional analyses would significantly strengthen the structural characterization of the polysaccharides and provide more comprehensive structural information.
Response: Concerning the limitations of using the selected in vitro assay to simulate the wound healing properties, we would like to thank you for the comments. Indeed we are aware about the limitations of such assays, however, the respective assay that monitors the migration of fibroblasts to close a scratch-made gap in the layer of cells, possibly is the most broadly used to simulate wound healing as “cell migration is a rate-limiting event during the wound-healing process to re-establish the integrity and normal function of tissue layers after injury” (DOI: 10.3791/56825). In vivo studies should follow to undoubtedly prove the efficacy of the specific eluate on wound healing. We have introduced such a suggestion in the discussion session, line 429-431.

Reviewer 2 Report
Comments and Suggestions for Authors
The work is based on the study of total polysaccharides and glucans extracted by an ultrasonic technique from 14 strains of microalgae and then on the study of polysaccharide composition by HPLC-DAD of two selected strains of Tetraselmis. The sulfated polysaccharides were separated by anion-exchange chromatography and tested for their different biological properties.The study seems properly conducted and carried out.
Please find below a list of comments and suggestions I hope can improve the quality of manuscript.
Different methods of extracting polysaccharides from microalgae and other organisms are described in the literature. In this study, the reasons why the authors selected ultrasound are unclear.
Consider replacing alpha and beta by their respective symbols in italic throughout the document.
Along the text, the standard deviation values following the various results are missing.
At the bottom of tables 2 and 3: Data represent mean ± SD with indication of the number of replicates.
Abstract: Dunaliella instead of Dunaliela
Line 90 and following - missing indication of what % represents (w/w?)
Line 97 and 98 – check the values; there is no agreement with those in table 1
Table 1 – check beta-glucans per 100 g of total sugars presented in lines 4 and 5
Line 165 - there is no need to repeat two equal values
Line 174 – Trolox equivalents
Table 3 – units of antioxidant capacity in Trolox equivalents
Lines 189, 203, 205 – Control assay/assays
Lines 477-483 – text is repeated
The incubation time of the cultures, kinetic growth parameters, final cell density of the culture are missing. Consider including culture growth curve in supplementary material.
Lines 492, 507- pH was adjusted to which value? How?
The reference for formula (1) is missing; the same for the one in line 511
Line 503 –What does the reagent blank consist of?
In the description of the fractionation, the volume of the stationary phase, column dimensions, brand of resin are missing.
Line 609 – 3 kDa
Line 631 – Trolox equivalents
For the MTT assay the description of background control is missing.
Please note that in the Material and Methods are several brands of the reagents missing, as well as for 96-well plates.
Author Response
Dear Reviewer,
I would like to thank you for your time and your comments, to which we shall endeavor to respond.
Different methods of extracting polysaccharides from microalgae and other organisms are described in the literature. In this study, the reasons why the authors selected ultrasound are unclear.
Response: Indeed, there are various extraction techniques, with hot water extraction being the most widely used, as well as methods such as ultrasound, CO2 explosion, or low-pressure homogenization, which rely on different types of equipment, etc. The choice of extraction using the combination of ultrasound and hot water was made based on international literature, as highlighted in the corresponding section (see 4.3.2. Microalgae cells’ disruption process), and the available equipment.
Consider replacing alpha and beta by their respective symbols in italic throughout the document.
Response: All ‘alpha’ and ‘beta’ were modified to ‘α’ and ‘β’ in italics throughout the manuscript, except for their first mention, where they were written out in full to explain the abbreviation.
Along the text, the standard deviation values following the various results are missing.
Response: Your observation is valid; however, this was done intentionally, as all mean values accompanied by the standard deviations are presented in. We consider it redundant to repeat the same information, as it may become tiresome for the reader. For any result of interest, the reader can refer to the corresponding table for complete information.
At the bottom of tables 2 and 3: Data represent mean ± SD with indication of the number of replicates.
Response: At the bottom of all three Tables, the recommended explanation was added.
Abstract: Dunaliella instead of Dunaliela
Response: Your observation is right; it has been properly corrected.
Line 90 and following - missing indication of what % represents (w/w?)
Response: Your observation is correct; it was an oversight on our part as we considered it self-evident. It has been properly corrected.
Line 97 and 98 – check the values; there is no agreement with those in table 1
Response: Your observation is correct; it was an unintentional error and has been properly corrected.
Table 1 – check beta-glucans per 100 g of total sugars presented in lines 4 and 5
Response: We checked them; in case of T. suecica ELO1-1, its 2.9/20.2=0.1435 gβ-glucans/gtotalsugars, or =14.4 gβ-glucans/100 gtotalsugars; in case of Tetraselmis spp. Mes5, its 1.0/20.1=0.0497 gβ-glucans/gtotalsugars, or =5.0 gβ-glucans/100 gtotalsugars
Line 165 - there is no need to repeat two equal values
Response: Your observation is correct; it has been properly corrected.
Line 174 – Trolox equivalents
Response: Your observation is correct; it has been properly corrected.
Table 3 – units of antioxidant capacity in Trolox equivalents
Response: It has been properly corrected.
Lines 189, 203, 205 – Control assay/assays
Response: It has been properly modified.
Lines 477-483 – text is repeated
Response: Your observation is right, it was an error that occurred during the transfer of the text into the manuscript format of the journal. It has been properly corrected.
The incubation time of the cultures, kinetic growth parameters, final cell density of the culture are missing. Consider including culture growth curve in supplementary material.
Response: The cultivation duration has been mentioned at the ‘Materials & Methods’ Section (L495), as recommended. For more information regarding (initial and final cell) density, dry biomass production values, μmax, etc., supplemental Table S1 was added, as recommended.
Lines 492, 507- pH was adjusted to which value? How?
Response: The pH was adjusted to the required values for the functioning of each enzyme, using the appropriate solutions following the manufacturer’s instructions. For further details, you may refer to the protocol of β-glucan enzyme assay kit (K-YBGL) of Megazyme® https://www.megazyme.com/beta-glucan-assay-kit-yeast-mushroom
The reference for formula (1) is missing; the same for the one in line 511
Response: For further details regarding the two formulas, you may refer to the protocol of β-glucan enzyme assay kit (K-YBGL) of Megazyme® https://www.megazyme.com/beta-glucan-assay-kit-yeast-mushroom
Line 503 –What does the blank consist of?
Response: The reagent blank solution contained, as is customary, all the components recommended by the manufacturer in the protocol, excluding the sample and replacing it with ddH₂O.
In the description of the fractionation, the volume of the stationary phase, column dimensions, brand of resin are missing.
Response: You are right, more information has been added at the text (L633-635).
Line 609 – 3 kDa
Response: It has been properly corrected.
Line 631 – Trolox equivalents
Response: It has been added.
For the MTT assay the description of background control is missing.
Response: You are right, more information has been added at the text (L709-711).
Please note that in the Material and Methods are several brands of the reagents missing, as well as for 96-well plates.
Response: You are right, ‘Materials and Methods’ Section has been enriched as recommended.

Reviewer 3 Report
Comments and Suggestions for Authors
1. Graphic figures are informative and easy to read.
2. The article covers the topics fully and extensively
3. The article will be of interest to a wide range of readers
4. Self-quoting is appropriate and ethical
5. The authors describe in detail the methods used to examine the samples, specifying the models of equipment used.
6. The findings of the authors of the article are in agreement with those of other authors.
Remarks:
1. Section 4 Materials and Methods
- In this study, microalgae biomass was freeze dried after cultivation, but the drying conditions and equipment model are not given by the authors.
- Aeration is used in the cultivation of microalgae biomass, so it is necessary to specify the process conditions.
- The cultivation of microalgae was carried out for how many days or to what optical density of the suspension?
- It is desirable to specify how long it took to grow what volume of biomass.
Section 4.3.2.
- How many cycles of centrifugation were performed until the molecules were completely extracted?
2. It is desirable to carry out FTIR studies of the obtained polysaccharides.
3. Lines 470-476 duplicate lines 477-48.
It is recommended to accept the article in print after minor revisions

Author Response
Dear Reviewer,
I would like to thank you for your time and your comments, to which we shall endeavor to respond immediately.
Remarks:
- Section 4 Materials and Methods
- In this study, microalgae biomass was freeze dried after cultivation, but the drying conditions and equipment model are not given by the authors.
Response: Specific information about freeze drying process has been added at the ‘Materials & Methods’ Section (L501-502), as recommended.
- Aeration is used in the cultivation of microalgae biomass, so it is necessary to specify the process conditions.
Response: Specific information about aeration process has been added at the ‘Materials & Methods’ Section (L497), as recommended.
- The cultivation of microalgae was carried out for how many days or to what optical density of the suspension?
Response: The cultivation duration has been mentioned at the ‘Materials & Methods’ Section (L495), as recommended. For more information regarding (initial and final cell) density, dry biomass production values, μmax, etc., supplemental Table S1 has been added, as recommended.
- It is desirable to specify how long it took to grow what volume of biomass.
Response: The duration has been mentioned at the ‘Materials & Methods’ Section (L495).
Section 4.3.2.
- How many cycles of centrifugation were performed until the molecules were completely extracted?
Response: The procedure was performed twice. I hope it is clearer now following the addition we made (Line 570).
- It is desirable to carry out FTIR studies of the obtained polysaccharides.
Response: Concerning the suggestion to use FTIR to further characterize the functional sulfate groups positioning on the anionic polysaccharides of T. verrucosa f. rubens PLA1-2 eluate E4, we had intentionally avoided this because we believed that such a detailed physicochemical characterization fell outside the scope of the study. The aim was to identify microalgal strains and eluates with bioactive properties, with the intention of addressing the physicochemical characterization of these polysaccharides in a subsequent publication. However, we proceeded with an FTIR analysis of this specific eluate at this manuscript (see ‘Materials & Methods’ – L672-681), as recommended. The results of the analysis and the spectrum are presented at Results Section – Subsection ‘2.3.’ (L179-198) and at supplemental Figure S1, respectively.
- Lines 470-476 duplicate lines 477-48.
Response: Your observation is right, it was an error that occurred during the transfer of the text into the manuscript format of the journal. It has been properly corrected.

Round 2
Reviewer 1 Report
Comments and Suggestions for Authors
Manuscript ID: marinedrugs-3425452
Wound healing, antioxidant, and antiviral properties of bioactive polysaccharides of microalgae strains isolated from Greek coastal lagoons
Gabriel Vasilakis et al.
Following the completion of the review and revision process, the author's thorough responses to this reviewer effectively addressed each of the points raised.
By incorporating FTIR analysis, you have provided valuable insights into the functional groups, significantly enhancing the characterization of the anionic polysaccharides and adding depth to the study. Additionally, the revision of Tables 2 and 3 to express hexose and sulfate content in molar ratios has clarified the degree of sulfation per sugar unit, thereby improving the overall comprehension of the chemical profiles. Furthermore, your acknowledgment of the limitations associated with the in vitro scratch assay, alongside your recommendations for further in vivo studies, demonstrates a clear understanding of these experimental constraints and sets the stage for future research developments.
I believe the revised version of the manuscript can be accepted in its present form.